# Skip-Graph: Learning graph embeddings with an encoder-decoder model

**John Boaz Lee & Xiangnan Kong**
Department of Computer Science
Worcester Polytechnic Institute
Worcester, MA 01609, USA
`{jtlee, xkong}@wpi.edu`

## Abstract

In this work, we study the problem of feature representation learning for graph-structured data. Many of the existing work in the area are task-specific and based on supervised techniques. We study a method for obtaining a generic feature representation for a graph using an unsupervised approach. The neural encoder-decoder model is a method that has been used in the natural language processing domain to learn feature representations of sentences. In our proposed approach, we train the encoder-decoder model to predict the random walk sequence of neighboring regions in a graph given a random walk along a particular region. The goal is to map subgraphs — as represented by their random walks — that are structurally and functionally similar to nearby locations in feature space. We evaluate the learned graph vectors using several real-world datasets on the graph classification task. The proposed model is able to achieve good results against state-of-the-art techniques.

## 1 Introduction

The skip-gram model (Mikolov et al., 2013) was originally introduced in the natural language processing (NLP) domain as a model for learning vector representations of words. Recently, it has been adapted successfully to solve the problem of learning node representations for graph-structured data (Grover & Leskovec, 2016; Perozzi et al., 2014). The learned vectors can then be used directly in problems such as link prediction (Miller et al., 2009), or clustering of nodes on a graph (Vinayak et al., 2014). However, in many real-world applications we need to learn a feature representation for the entire graph instead of representations for just the nodes in the graph. In this paper, we study the graph representation learning problem, where the task is to learn a feature representation for any graph object. We propose a novel solution based upon the encoder-decoder model.

Graph-structured data can be found in many different domains including biology, chemistry, and the study of social networks. For instance, in chemistry, chemical compounds can be represented as molecular graphs (Duvenaud et al., 2015). In social network analysis, the interaction among different entities of a community can be captured using a social graph (Yanardag & Vishwanathan, 2015). A natural question that arises in these scenarios is what the structure of a graph tells us about the properties of the graph (*e.g.,* what does the molecular graph tell us about the compound's aqueous solubility, or its anti-cancer activity?). In other words, we are often interested in performing machine learning tasks on graph-structured data. Many techniques have been proposed to solve this problem, these include learning graph kernels (Vishwanathan et al., 2010), identifying discriminative subgraphs (Kong et al., 2011), using specially designed neural network models such as the graph neural network (Scarselli et al., 2009), and learning the graph fingerprint (Duvenaud et al., 2015). Most of the approaches for learning graph features are supervised and task-specific. Our approach, on the other hand, is unsupervised and general-purpose. The learned features can be used directly with off-the-shelf machine learning methods on different tasks, such as classification or clustering. Perhaps the work that resembles this work the most is the one in (Yanardag & Vishwanathan, 2015). We argue, however, that our approach is different and this is good motivation to pursue the study as there has not been many work published in the area. For one, we use the skip-thought model (Kiros

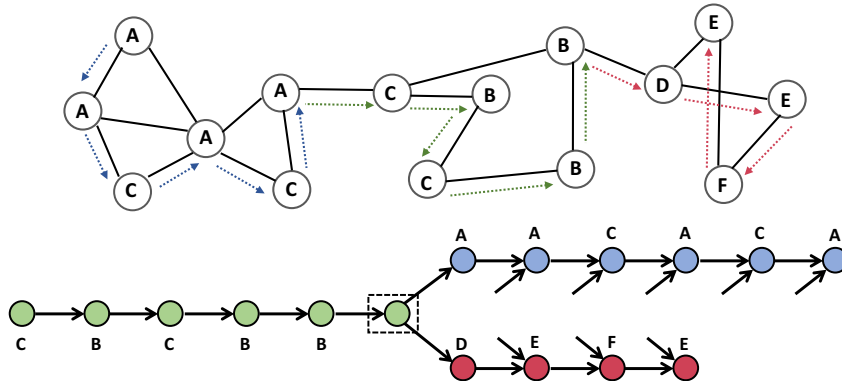

Figure 1: A random walk over a graph is split into three subsequences $(s_1, s_2, s_3)$. The middle sequence is input into the encoder and the decoders attempt to reconstruct the previous and next sub-sequence. The unattached arrows are connected to the encoder output to condition the decoder.

et al., 2015) and we are not just interested in structurally similar subgraphs but also functionally similar ones.

Our approach is based on the encoder-decoder model (Kalchbrenner & Blunsom, 2013; Cho et al., 2014); in particular, we are interested in the skip-thought model. In (Kiros et al., 2015), tuples composed of three consecutive sentences from word documents are fed into an RNN model and the model attempts to reconstruct the previous and next statements given the middle sentence. After training on a large text corpus, the hidden vector values for an input sentence can be used as that input sequence's feature representation. It has been shown that the model learns a function that maps semantically and syntactically similar sentences close to one another in feature space. In this work, the idea is to take instead a sequence generated by a random walk along a labeled graph and to divide it into three parts, feeding these into the encoder-decoder model. Since the structure of the graph determines the random walk sequences that can be generated, we can treat each sub-sequence as a representation of a particular subgraph in the graph. We argue that by training an encoder-decoder model on a large number of random walk sequences, we can learn a feature representation that groups structurally and functionally similar subgraphs together. Figure 1 shows an example of how we can train the model using a random walk over a graph. A simple example that illustrates how the model may learn to identify functionally similar subgraphs is shown in Figure 2.

After the model is trained on a large sample of random walks generated from a dataset of labeled graphs, we can then freeze the model and use the encoder as a feature extractor. In particular, we obtain a feature representation of a graph by sampling multiple short random walks and aggregating the information encoded in the feature representations of these short walks. We borrow an analogy from the NLP domain to highlight the idea. In order to obtain a good feature representation for a text document, short of sampling all the words in the document one may sample a set of sentences from the document and use these to construct the features for the document. Similarly, to obtain a feature representation for a graph, we sample a set of subgraphs (as represented by the short walks) and use the aggregate subgraph features to construct the final graph feature vector. Since we use the trained encoder as our feature extractor, graphs that share structural and functional properties will tend to have more similar feature vectors.

## 2 PROPOSED METHOD

### 2.1 SKIP-THOUGHT

Since our proposed approach is based on the encoder-decoder model of (Kiros et al., 2015), we begin by briefly introducing the model. The encoder-decoder model uses an RNN with GRU (Chung et al., 2014) activation as the encoder and an RNN with a conditional GRU as the decoder. The model is trained using the Adam stochastic optimization algorithm (Kingma & Ba, 2015).

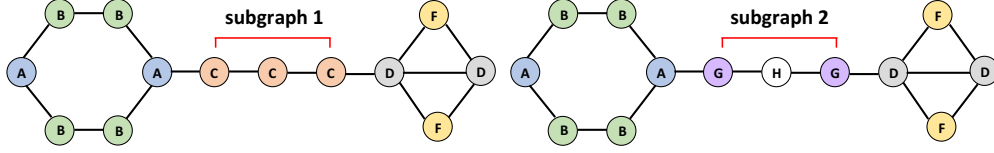

**possible random walk sequences: "B-B-A-B-B-A-C-C-C-D-F-D-F", "B-B-A-B-B-A-G-H-G-D-F-D-F"**

Figure 2: Two structurally dissimilar subgraphs can be considered functionally similar if they always appear in the same neighborhood. For instance, subgraphs "C-C-C" and "G-H-G" are structurally different since they are composed of different types of nodes but they seem to be serving the same function of connecting the same kind of regions together. If these patterns appear frequently in the dataset, the encoder-decoder model will learn very similar representations for the random walk sequences corresponding to the two subgraphs.

The input to the model is a tuple of sentences $(s_{i-1}, s_i, s_{i+1})$, with $x_i^t$ being the word embedding for the $t$-th word, $w_i^t$, of sentence $s_i$. The word embeddings for the middle sentence, $s_i$, are fed sequentially as input to the the encoder. The encoder generates a hidden vector $h_i^t$ at each time step $t$, this is the information the model retained after processing sequence $x_i^1$, ... , $x_i^t$ and can be thought of as the sequence representation. The hidden state $h_i^N$ can thus be considered the sentence representation, given $s_i$ is of length $N$. Given a sequence to encode, the encoder iterates through the following equations, as given in (Kiros et al., 2015). Here the subscripts $i$ are dropped for simplicity.

$$\mathbf{r}^t = \sigma(\mathbf{W}_r\mathbf{x}^t + \mathbf{U}_r\mathbf{h}^{t-1}) \tag{1}$$

$$\mathbf{z}^t = \sigma(\mathbf{W}_z\mathbf{x}^t + \mathbf{U}_z\mathbf{h}^{t-1}) \tag{2}$$

$$\bar{\mathbf{h}}^t = \tanh(\mathbf{W}\mathbf{x}^t + \mathbf{U}(\mathbf{r}^t \odot \mathbf{h}^{t-1})) \tag{3}$$

$$\mathbf{h}^t = (1 - \mathbf{z}^t) \odot \mathbf{h}^{t-1} + \mathbf{z}^t \odot \bar{\mathbf{h}}^t \tag{4}$$

where $\mathbf{r}^t$ is the forget gate, $\mathbf{z}^t$ is the update gate, $\bar{\mathbf{h}}^t$ is the proposed hidden state, and $\odot$ is the component-wise product. Here $\mathbf{r}^t$ decides what information to discard from the previous state, $\mathbf{z}^t$ decides what new information to encode, and the new hidden vector $\mathbf{h}^t$ is calculated accordingly. Values in $\mathbf{r}^t$ and $\mathbf{z}^t$ are in the range $[0, 1]$.

Two decoders with separate parameters are used to reconstruct the previous statement $s_{i-1}$ and the next statement $s_{i+1}$. The computation for the decoder is similar to that of the encoder, except this time the models are also conditioned on the encoder output $\mathbf{h}_i$. Decoding involves iterating through the following statements. Again the subscript $i + 1$ (similarly, $i - 1$) is dropped.

$$\mathbf{r}^t = \sigma(\mathbf{W}_r^d\mathbf{x}^{t-1} + \mathbf{U}_r^d\mathbf{h}^{t-1} + \mathbf{C}_r\mathbf{h}_i) \tag{5}$$

$$\mathbf{z}^t = \sigma(\mathbf{W}_z^d\mathbf{x}^{t-1} + \mathbf{U}_z^d\mathbf{h}^{t-1} + \mathbf{C}_z\mathbf{h}_i) \tag{6}$$

$$\bar{\mathbf{h}}^t = \tanh(\mathbf{W}^d\mathbf{x}^{t-1} + \mathbf{U}^d(\mathbf{r}^t \odot \mathbf{h}^{t-1}) + \mathbf{C}\mathbf{h}_i) \tag{7}$$

$$\mathbf{h}_{i+1}^t = (1 - \mathbf{z}^t) \odot \mathbf{h}^{t-1} + \mathbf{z}^t \odot \bar{\mathbf{h}}^t \tag{8}$$

here the $\mathbf{C}$ matrices are used to bias the computation by the sentence vector produced by the encoder. Also, note that the word embeddings are from the previous and next statements since these are what is given to the decoders. The probability of word $w_{i+1}^t$ can be calculated by

$$P(w_{i+1}^t|w_{i+1}^{<t}, \mathbf{h}_i) \propto \exp(\mathbf{v}_{w_{i+1}^t}\mathbf{h}_{i+1}^t) \tag{9}$$

where $\mathbf{v}_{w_{i+1}^t}$ is the row vector in the vocabulary vector $\mathbf{V}$ corresponding to the word $w_{i+1}^t$. The vocabulary matrix, $\mathbf{V}$, is a weight matrix shared by both decoders connecting the decoder's hidden states for computing a distribution over words.

Finally, given a sentence tuple, the training objective is given by

$$\sum_t \log P(w_{i+1}^t|w_{i+1}^{<t}, \mathbf{h}_i) + \sum_t \log P(w_{i-1}^t|w_{i-1}^{<t}, \mathbf{h}_i) \tag{10}$$

which is the sum of log-probabilities for the words in the previous and next statements, $s_{i-1}$ and $s_{i+1}$, conditioned on the sentence representation for $s_i$. The total objective would then be the above summed for all tuples in the training data.

## 2.2 Skip-graph

In this work, we are interested in graph-structured data in particular. In our setting, we are given a set of labeled graphs $\mathcal{D} = \{\mathcal{G}_1, \mathcal{G}_2, \ldots, \mathcal{G}_n\}$ with each graph associated with a class label. A graph $\mathcal{G} = (\mathcal{V}, \mathcal{E}, \ell_v)$ is comprised of a vertex set $\mathcal{V}$, an edge set $\mathcal{E} \subseteq \mathcal{V} \times \mathcal{V}$, and a node labeling function $\ell_v : \mathcal{V} \to \mathcal{L}_{\mathcal{V}}$ which assigns each node to a label in $\mathcal{L}_{\mathcal{V}}$. Additionally, the edges may also be labeled in which case we also have an edge labeling function $\ell_e : \mathcal{E} \to \mathcal{L}_{\mathcal{E}}$. Nodes and edges can also have associated feature vectors, these are $\mathbf{f}_v \in \mathbb{R}^{D_v}$, and $\mathbf{f}_e \in \mathbb{R}^{D_e}$, respectively.

### 2.2.1 Unlabaled graphs

Although we will be working primarily with labeled graphs, our method can be easily extended to support unlabeled graphs by including an additional pre-processing step. Algorithms like the Weisfeiler-Lehman algorithm (Weisfeiler & Lehman, 1968; Shervashidze et al., 2011) or the Morgan algorithm (Rogers & Hahn, 2010) for calculating molecular fingerprints are iterative algorithms that work by repeatedly calculating the attribute for a node via hashing of the attributes of its neighboring nodes. The final node attributes capture the local structure or topology of the graph. For unlabeled graphs, all node attributes can be initialized to a constant value and after the algorithm is run, we can treat the node attributes as the labels for the nodes in the graph.

### 2.2.2 Training set generation

Given a set of graphs $\mathcal{D}$, a sample size $K$, a minimum random walk length $l_{min}$, and a maximum random walk length $l_{max}$, we take each graph $\mathcal{G} \in \mathcal{D}$ and generate $K$ random walk sequences. Specifically, for a graph $\mathcal{G}$, $K$ sequences of the form

$$\ell_v(v_1), \ldots, \ell_v(v_k), \ell_v(v_{k+1}), \ldots, \ell_v(v_{k+k\prime}), \ell_v(v_{k+k\prime+1}), \ldots, \ell_v(v_{k+k\prime+k\prime\prime})$$

are generated. Here, $v_1 \in \mathcal{V}$ is a randomly selected start node, $(v_i, v_{i+1}) \in \mathcal{E}$ for $i$ from $1 \ldots k + k\prime + k\prime\prime - 1$, and $l_{min} \geq k, k\prime, k\prime\prime \geq l_{max}$. We can split each sequence into three sub-sequences with $s_1 = \ell_v(v_1), \ldots, \ell_v(v_k)$, $s_2 = \ell_v(v_{k+1}), \ldots, \ell_v(v_{k+k\prime})$, and $s_3 = \ell_v(v_{k+k\prime+1}), \ldots, \ell_v(v_{k+k\prime+k\prime\prime})$. For each sequence, $k, k\prime$, and $k\prime\prime$ are randomly drawn to be between the constraints. Since the length of the sub-sequences do not need to have fixed lengths and can instead be between $l_{min}$ and $l_{max}$, regions of varying sizes can easily be considered.

In the above formulation, we assume that only the vertices in the graph are labeled and node and edge features are not given. When nodes, or edges, are labeled and feature vectors are provided we can use a one-hot embedding to represent each unique combination of labels and features. This treats each distinct combination as a unique "word" and does not capture the relationship between nodes or edges that share labels or certain features. A better approach is to simply use a one-of-$|\mathcal{L}|$ vector to encode the label and concatenate this with the feature vector, this allows the node or edge embedding to capture shared features and labels.

Once all the tuples of random walk sequences have been generated, they can be used to train the encoder-decoder[1] in an unsupervised fashion.

### 2.2.3 Obtaining final graph representation

After the encoder-decoder has been trained, we can freeze the model and use the encoder to generate representations, $\mathbf{h}_i$, for any arbitrary random walk sequence. Ultimately, however, we are interested in obtaining a representation for entire graphs so we try several strategies for aggregating the encoder representations obtained from a set of independent random walks sampled from a given graph.

1. **Single walk:** In this approach we do not use several encoder representations. Instead, we train the model on relatively long (relative to the size of the graphs in the dataset) random walk sequences and use a single long walk over the graph to obtain its representation.

2. **Average:** We compute the component-wise average of the encoder representations of the sampled random walk sequences. This is then used as the graph representation.

---

[1]We use the implementation in https://github.com/ryankiros/skip-thoughts.

3. **Max:** As in (Kiela & Bottou, 2014), we take the component-wise absolute maximum of all encoder representations.

4. **Cluster:** The encoder representations are first fed into a clustering technique like K-means (Hamerly & Elkan, 2003) and we use the cluster information to create a bag-of-cluster vector that serves as the graph's representation.

The procedure for obtaining the graph embeddings is summarized in Algorithm 1. The calculated graph embeddings can now be used with any off-the-shelf machine learning method.

---

**Algorithm 1:** Calculate graph embedding

**Input** : Training set $\mathcal{D}$, sample size $K$, walk lengths $l_{min}$ and $l_{max}$, aggregate sample size $K'$, and aggregate method $agg$

**Output**: Graph embeddings

1 Generate set of $K \times |\mathcal{D}|$ random walk tuples, $\mathcal{S}$;
2 Train encoder-decoder model using $\mathcal{S}$;
3 **for** *each $\mathcal{G}$ in $\mathcal{D}$* **do**
4 Randomly select $K'$ random walks;
5 Obtain encoder representations $\mathbf{h}_1, ..., \mathbf{h}_{K'}$ from the random walks;
6 Compute graph embedding with $agg(\mathbf{h}_1, ..., \mathbf{h}_{K'})$;
7 **end**
8 Return final graph embeddings;

---

## 3 EXPERIMENTS

### 3.1 DATASET

We evaluate our proposed method on the binary classification task using four chemical compound datasets (Kong et al., 2011). The datasets contain chemical compounds encoded in the SMILES format (Weininger, 1988); class labels indicate the anti-cancer properties (active or inactive) of each compound. We use the RDKit[2] package to obtain the molecular graphs from the SMILES data. We also use RDKit to obtain the labels for the nodes (atom type) and edges (bond type). Additionally, we used the number of attached hydrogens as a node feature and bond conjugation as an edge feature. Since the edges in the datasets we evaluate on are also labeled, the generated random walk sequences include edges. The datasets are all highly skewed with far more negative samples than positive ones, we tested the methods on balanced datasets by selecting a random set of negative samples equal to the positive ones. Table 1 shows a summary of the datasets used. The average size of the molecular graphs in each of the four datasets is around 30.

Table 1: Summary of experimental datasets. "# pos" stands for the number of positive samples.

| dataset | # graphs | # pos | details |
|---------|----------|-------|---------|
| NCI81 | 40700 | 1396 | Colon Cancer |
| NCI83 | 27992 | 2276 | Breast Cancer |
| NCI123 | 40152 | 3112 | Leukemia |
| HIV | 7781 | 266 | HIV Anti-virus |

### 3.2 COMPARED METHODS

We compared our proposed approach with several state-of-the-art techniques. Since the method is a task-irrelevant way to obtain graph representations, the goal of the paper isn't necessarily to come up with a method that achieves absolute best performance on the tested datasets so we do not test against an exhaustive list of methods. Our primary objective is to see whether the method can

---

[2]http://www.rdkit.org/

potentially be used to learn useful graph embeddings as a starting point for future investigation in the area. Since we are testing the method using molecular graph datasets, we chose to compare against techniques that have achieved state-of-the-art performance on these type of graphs. We also compare against a method that learns node embeddings instead of an entire graph embedding. The tested methods are:

- **ECFP** (Rogers & Hahn, 2010): Extended-connectivity circular fingerprints, which are a refinement of the Morgan algorithm (Morgan, 1965), use an iterative approach to encode information about substructures in a molecular graph in a fingerprint vector. In this method a hash function is used to map the concatenated features from a neighborhood to an index in the fingerprint vector.

- **NeuralFPS** (Duvenaud et al., 2015): Neural fingerprints replace the function that is used to compute a fingerprint vector with a differentiable neural network. This allows the method to learn from the data, prioritizing useful or discriminative features.

- **DeepWalk** (Perozzi et al., 2014): The DeepWalk model learns representations for nodes in a single graph. However, we can also train the model using random walks from multiple graphs if the various graphs share the same kind of nodes. The model will then learn to generate similar representations for nodes that co-occur frequently across all the graphs. To generate the final embedding for a graph, we can simply apply average pooling to the vectors of all the nodes in the graph – which is a reasonable strategy to capture the overall profile of the graph.

- **Skip-graph**: Our proposed method. We train an encoder-decoder model using random walks generated from the graphs and use the encoder's random walk representation to calculate the graph embedding.

To test ECFP and NeuralFPS, we used the library[3] provided by (Duvenaud et al., 2015). The size of the graph embedding was restricted to 164 for all methods and a grid-search was done to optimize the parameters of the various methods. For ECFP and NeuralFPS, we tested different values for the following parameters: fingerprint radius, $\ell_2$ regularization penalty, step size for the optimization, hidden layer dimension, and convolution layer dimension (only for NeuralFPS). All results reported are the average over 5-fold cross validation. Since a neural network, with a single hidden layer, was used as the classifier in Duvenaud et al. (2015), we chose to use the same classifier for our model and the grid-search was performed over the same set of values for classifier-related parameters. In particular, for the neural network, we tested various settings with hidden layer size selected from $\{70, 100, 140\}$, and $\ell_2$ regularization chosen from $\{0.0001, 0.001, 0.01, 0.1\}$.

## 3.3 CLASSIFICATION RESULTS

We show the classification accuracy of the different methods in Table 2. The proposed method achieves top performance in three of the four datasets we tested. It is a little surprising, however, to find that NeuralFPS performs slightly worse than ECFP. This seems to suggest that it is overfitting the data as NeuralFPS is a generalization of ECFP and should, in theory, be at least as good as ECFP. Also, we find that averaging the DeepWalk embeddings trained from random walks generated from the entire training set can be a simple yet effective way to generate a graph representation.

Table 2: Summary of experimental results.

| method | dataset | | | |
|---|---|---|---|---|
| | HIV | NCI81 | NCI83 | NCI123 |
| ECFP | 68.30% | 68.90% | 62.06% | 60.17% |
| NeuralFPS | 67.48% | 65.24% | 59.91% | 60.00% |
| DeepWalk | 69.90% | 68.00% | **63.89%** | **64.43%** |
| Skip-graph | **72.77%** | **69.98%** | **63.80%** | 62.60% |

---

[3]https://github.com/HIPS/neural-fingerprint

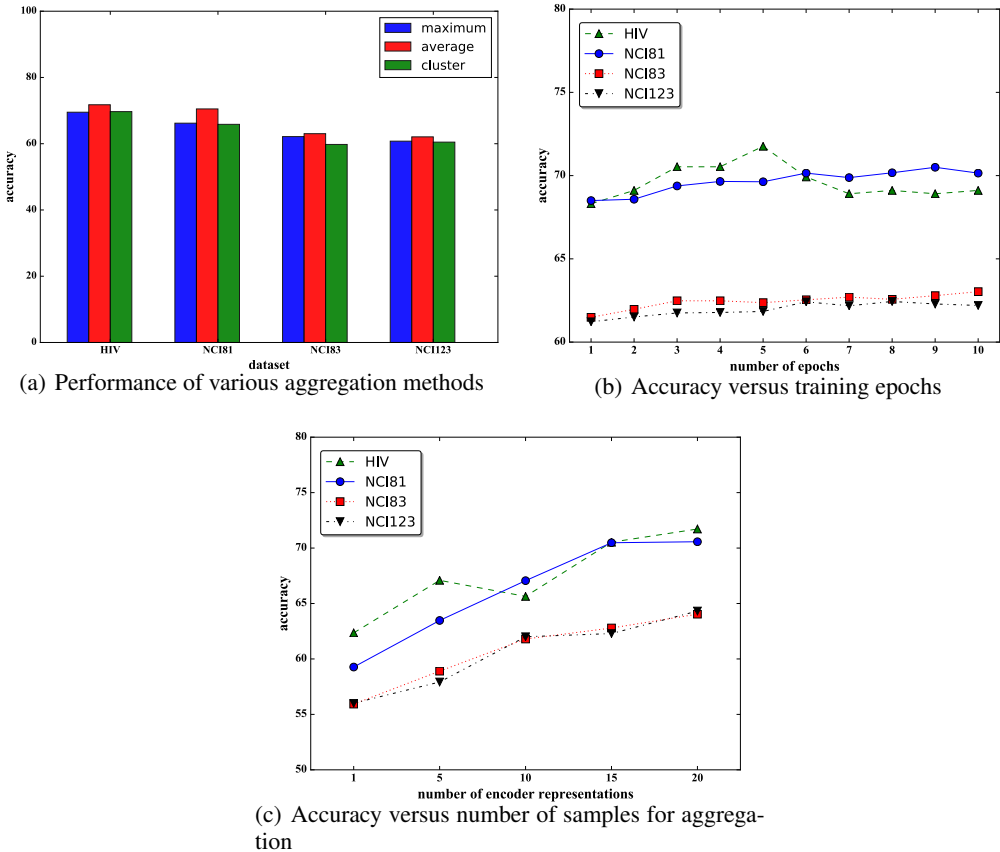

(a) Performance of various aggregation methods

(b) Accuracy versus training epochs

(c) Accuracy versus number of samples for aggregation

Figure 3: The performance of our proposed method under various settings.

## 3.4 PARAMETER STUDY

We tested the performance of the method using the various aggregation methods. The performance was extremely poor when we trained the encoder-decoder model on long random walks and used a single long walk to generate the graph representation. The other three aggregation strategies yielded better results. Figure 3(a) shows the performance of these methods. Averaging the hidden vector representations seems to yield the best performance, calculating the component-wise maximum yielded the second best results while the method that had the additional cluster pre-processing step performed slightly worse.

We plot the accuracy of the method over the number of training epochs in Figure 3(b). With the exception of the HIV dataset, which has a relatively few number of samples, the results show a gradual increase in the classification accuracy as the number of training epochs is increased. This is consistent with results in other work that show that given a large number of training data, recurrent neural models generally achieve better results when trained longer.

Figure 3(c) shows the accuracy in the classification task over different sample sizes $K'$, or the number of samples aggregated to obtain the final graph representation. It is clear from the results that a better graph representation is obtained if we use more samples to calculate the final graph representation. This is quite intuitive as a limited sample may not be representative and may fail to capture the properties of the graph well enough.

We tested several different values for $l_{min}$ and $l_{max}$ and the one that seemed to perform best in our case was $l_{min} = 7$ and $l_{max} = 12$. This is a reasonable constraint on the random walk length given that the average size of the molecular graphs was around 30. We used $K = 100$ when generating a set of random walks to train the encoder-decoder.

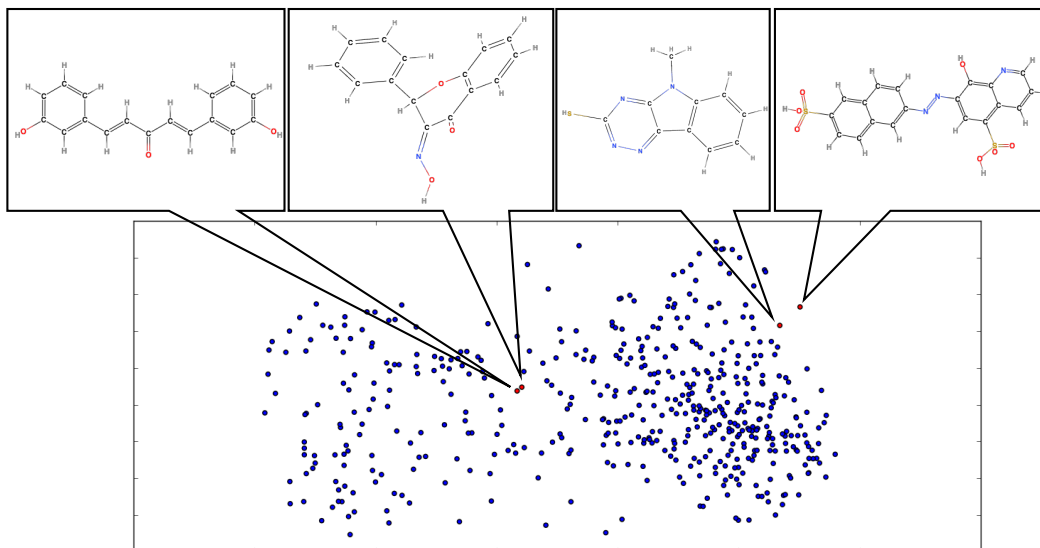

Figure 4: The learned embeddings for graphs in the HIV dataset. The 2-d representations were calculated using Kernel PCA (Mika et al., 1998).

### 3.5 VISUALIZATION OF GRAPH EMBEDDINGS

We show a scatterplot of the HIV graph embeddings learned by our model in Figure 4. In particular, we highlight two pairs of graphs that had very similar embeddings. We note that the first pair of graphs (the one on the right) are structurally similar, that is they have a large sub-structure in common. The graphs in the second pair each contain two similar substructures that are joined by segments that appear to be "functionally" similar.

### 3.6 USING AN ENSEMBLE OF CLASSIFIERS

Since it is possible to generate many different sets of random walks to train the encoder-decoder model, we tried training five encoders on five separate sets of random walks. An ensemble (Opitz & Maclin, 1999) of five classifiers is then created with each classifier trained on the graph representations obtained from one of the five encoders. We compare the predictive accuracy of the ensemble versus the single classifier when all other settings are fixed. We observed a slight improvement (around $1 - 3\%$) in the accuracy of the model. All the results reported above are for the single classifier case.

## 4 CONCLUSION

We introduced an unsupervised method, based on the encoder-decoder model, for generating feature representations for graph-structured data. The model was evaluated on the binary classification task on several real-world datasets. The method outperformed several state-of-the-art algorithms on the tested datasets.

There are several interesting directions for future work. For instance, we can try training multiple encoders on random walks generated using very different neighborhood selection strategies. This may allow the different encoders to capture different properties in the graphs. We would also like to test the approach using different neural network architectures. Finally, it would be interesting to test the method on other types of heterogeneous information networks.

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
