# Peer review of "Skip-graph: Learning graph embeddings with an encoder-decoder model"

_ICLR 2017 — rejected_

[Public Comment · (anonymous) · 05 Nov 2016]
**Baseline comparisons**

Hi,

In the abstract you have mentioned, "Many of the existing work in the area are task-specific and based on supervised techniques". I don't think this is a valid statement, since lot of recent work in graph embeddings is based on unsupervised methods. Like LINE and DeepWalk. 

And is there any particular reason you didn't compare your method with DeepWalk and LINE, considering the fact that they are state-of-the art methods in representation learning for graphs?

Regards,

[Official Review · AnonReviewer2 · rating 7 · confidence 3 · 16 Dec 2016]
**Good paper**

The paper presents a method to learn graph embeddings in a unsupervised way using random walks. It is well written and the execution appears quite accurate. The area of learning whole graph representations does not seem to be very well explored in general, and the proposed approach enjoys having very few competitors.

In a nutshell, the idea is to linearize the graph using random walks and to compute the embedding of the central segment of each walk using the skip-thought criterion. Being not an expert in biology, I can not comment whether or not this makes sense, but the gains reported in Table 2 are quite significant. 

An anonymous public comment compared this work to a number of others in which the problem of learning representations of nodes is considered. While this is arguably a different goal, one natural baseline would be to pool these representations using mean- or max- pooling. It would very interesting to do such a comparison, especially given that the considered approach heavily relies on pooling (see Figure 3(c))

To sum up, I think it is a nice paper, and with more baselines I would be ready to further increase the numerical score.

[Official Review · AnonReviewer3 · rating 6 · confidence 1 · 19 Dec 2016]
**An extension of skip-graph architecture to classifying similar molecular graphs**

Authors take the skip-graph architecture (Kiros 2015) and apply it to classifying labeled graphs (molecular graphs). They do it by creating many sentences by walking the graph randomly, and asking the model to predict previous part and next part from the middle part. Activations of the decoder part of this model on a walk generated from a new graph are used as features for a binary classifier use to predict whether the molecule has anti-cancer properties.

Paper is well written, except that evaluation section is missing details of how the embedding is used for actual classification (ie, what classifier is used)

Unfortunately I'm not familiar with the dataset and how hard it is to achieve the results they demonstrate, that would be the important factor to weight on the papers acceptance.

[Official Review · AnonReviewer1 · rating 5 · confidence 4 · 29 Dec 2016]
**Comparison with Graph kernels Missing**

This paper studies the graph embedding problem by using the encoder-decoder method. The experimental study on real network data sets show the features extracted by the proposed model is good for classification.

Strong points of this paper:
  1. The idea of using the methods from natural language processing to graph mining is quite interesting.
  2. The organization of the paper is clear

Weak points of this paper:
  1. Comparisons with state-of-art methods (Graph Kernels) is missing. 
  2. The problem is not well motivated, are there any application of this. What is the different from the graph kernel methods? The comparison with graph kernel is missing. 
  3. Need more experiment to demonstrate the power of their feature extraction methods. (Clustering, Search, Prediction etc.)
  4. Presentation of the paper is weak. There are lots of typos and unclear statements. 
  5. The author mentioned about the graph kernel things, but in the experiment they didn't compare them. Also, only compare the classification accuracy by using the proposed method is not enough.

[Public Comment · (anonymous) · rating 5 · confidence 4 · 02 Jan 2017]
**Experiments Are Incomplete**

This paper proposes an unsupervised graph embedding learning method based on random walk and skip-thought model. They show promising results compared to several competitors on four chemical compound datasets.

Strength:
1, The idea of learning the graph embedding by applying skip-thought model to random walk sequences is interesting. 
2, The paper is well organized.

Weakness:
1, As the current datasets are small (e.g., the average number of nodes per graph is around 30), it would be great to explore larger graph datasets to further investigate the method. 
2, Comparisons with recent work like LINE and node2vec are missing. You can compare them easily by applying the same aggregation strategy to their node embeddings.

Detailed Questions:
1, The description about how to split the random walk sequence into 3 sub-sequences is missing. Also, the line “l_min >= (n_k - 1), … >= l_max” in section 2.2.2 is a mistake.
2, Can you provide the standard deviations of the 5-fold cross validation in Table 2? I’m curious about how stable the algorithm is.

[Author Response · John Boaz Lee · 11 Jan 2017]
**Summary of changes**

Dear reviewers, thank you very much for all the insightful comments and suggestions. Please find below a summary of our response to each of the points that were raised. Please note that we have paraphrased some of the comments and combined similar ones.

Anonymous Reviewer 2:

Comment 1. Use the same classifier for all the compared methods.

We have updated the results in our paper to reflect the changes from using a common classifier for all the compared methods. The changes can be found in section 3.2.

Comment 2. Use mean- or max-pooling to aggregate the embeddings learned by methods such as DeepWalk or node2vec and include this as a baseline.

The paper has been updated to include results from Deepwalk which was added as an additional baseline. Changes were made to sections 3.2 and 3.3.

Anonymous Reviewer 3:

Comment 3. Provide more information about the exact values of the parameters that were used in the grid search.

The information have been added to the paper. This can be found in section 3.2.

Anonymous Reviewer 1:

Comment 4. Need more experiment to demonstrate the power of their feature extraction methods. (Clustering, Search, Prediction etc.). Only comparing the classification accuracy by using the proposed method is not enough.

We plan to extend the experiments by using the embedding for graph clustering and graph search tasks in future works/journal extension. Due to the suggested page limit of ICLR, it is not easy to add these additional experiments without deleting some existing results in the paper. In the existing results, we have demonstrated the power of the embedding through prediction tasks and visualization of the embedding.

Comment 5. Presentation of the paper is weak. There are lots of typos and unclear statements. 

The latest version of the paper has been proofread by several individuals and to the best of our knowledge we have removed the major typos and/or unclear statements. 

Comment 6. What is the difference from graph kernel methods? The comparison with graph kernel is missing. 

We agree that the problem studied in the deep graph kernel paper seems to be very similar, or even the same, to the one we are studying. However, the underlying approach is slightly different and we argue that that in itself is a good motivation for the work as there has not been too many work published in the area. We use an encoder-decoder model which has not been used, and for one we can identify functionally similar subgraphs. 

Regrettably, due to time constraints we are unable to add a comparison with Deep Graph Kernels. However, we have tested against methods that have been shown to achieve good results on the type of graphs we used (ECFP and NeuralFPS) and we have also tested against DeepWalk. The introduction section has been updated to include some discussion on this.

Anonymous Reviewer 4:

Comment 7. Comparisons with recent work like LINE and node2vec are missing. You can compare them easily by applying the same aggregation strategy to their node embeddings.

We have already addressed the comment of a previous reviewer and have included DeepWalk as a baseline. We did not add comparisons against node2vec and LINE as these, in general, belong to the same family of methods.

Comment 8. As the current datasets are small (e.g., the average number of nodes per graph is around 30), it would be great to explore larger graph datasets to further investigate the method. 

To compensate in a way, we tested the method on four different molecular graph datasets. Regrettably, we are unable to add experiments on more datasets at the moment.

Comment 9. The description about how to split the random walk sequence into 3 sub-sequences is missing. Also, the line “l_min >= (n_k - 1), … >= l_max” in section 2.2.2 is a mistake.

Section 2.2.2 has been updated to correct this. Thank you very much for pointing this out.

[Final Decision · Program Chairs · 06 Feb 2017]
**ICLR committee final decision**

The idea of applying skip-graphs to this graph domain to learn embeddings is good. The results demonstrate that this approach is competitive, but do not show a clear advantage. This is difficult, as the variety of approaches in this area is rapidly increasing. But comparisons to other methods could be improved, notably deep graph kernels.